# Effects of Different-Sized Cages on the Production Performance, Serum Parameters, and Caecal Microbiota Composition of Laying Hens

**DOI:** 10.3390/ani13020266

**Published:** 2023-01-12

**Authors:** Yi Wan, Qiang Du, Duobiao Wang, Ruiyu Ma, Renrong Qi, Rongbin Yang, Xin Li, Junying Li, Wei Liu, Yan Li, Kai Zhan

**Affiliations:** 1Anhui Key Laboratory of Livestock and Poultry Product Safety Engineering, Institute of Animal Husbandry and Veterinary Medicine, Anhui Academy of Agricultural Sciences, Hefei 230031, China; 2Anhui Sundaily Village Ecological Food Co., Ltd., Tongling 244100, China; 3Anhui Biaowang Farming Co., Ltd., Huaibei 235100, China

**Keywords:** cage size, laying hen, production performance, serum parameter, caecal microbiota composition

## Abstract

**Simple Summary:**

Cage size is an essential environmental factor affecting the well-being and performance of hens and thus it is necessary to determine the extent of its contribution. The present study investigated the effects of four different-sized cages, i.e., huge, large, medium, and small cages, on the productive performance, serum biochemical indices, and caecal microbiota composition of Roman laying hens. The results indicated that enlarging cage size has some positive effects on the productive performance, lipid metabolism, and antioxidant capacity of laying hens, while medium cages were superior to large cages in feed conversion efficiency and caecal microflora composition.

**Abstract:**

The effects of four different-sized cages—huge (HC), large (LC), medium (MC), and small (SC) cages—on the productive performance, serum biochemical indices, and caecal microbiota composition of Roman laying hens were investigated. At 44 weeks of age, a total of 450 hens were selected and allocated to the four groups, with six replicates each. Equal stocking density (0.054 m^2^ per bird) was maintained among the four groups throughout the experiment, and number of birds/cage changed for each treatment. After 2 weeks of preliminary trial, the formal experiment was performed from 46 to 60 weeks of age. The laying rate and feed conversion ratio (FCR) were determined daily, antibody titres were measured every 3 weeks, and serum biochemical parameters and caecal microbiota composition were analysed at 60 weeks of age. Compared to HC and SC, the higher laying rate and lower FCR in MC and LC indicated positive effects on egg production and feed efficiency, while SC showed the highest body weight gain (*p* < 0.05). With increasing cage size, the serum triglycerides (TG) and total cholesterol (T-CH) levels were reduced, and serum glutathione peroxidase (GSH-Px) activity improved, where birds raised in HCs had the lowest serum TG and T-CH and the highest GSH-Px activity. Twenty-nine different phyla and 301 different genera were detected in the caecal microbiota of birds in the four groups. *Methanobrevibacter* was significantly higher in the SC than in the other groups (*p* < 0.05). *Faecalibacterium* was most abundant in the MC compared with the other groups (*p* < 0.05) and was significantly positively correlated with serum GSH-Px concentration (*R* = 0.214, *p* = 0.0017). *Lactobacillus* was significantly less abundant in the LC and MC than in the HC and SC groups (*p* < 0.05) and was significantly positively correlated with body weight (*R* = 0.350, *p* = 0.0009) but negatively correlated with laying rate and FCR. In conclusion, MC were superior to HC and LC in improving feed conversion efficiency and caecal microflora composition compared to the SC. An appropriate increase in cage size is beneficial to laying hen production and health.

## 1. Introduction

With the increased interest in humane care aimed at poultry welfare, much attention has been given to the rearing conditions and space allowance of hens. Researchers continue to seek methods for improving the well-being of laying hens and evaluating the outcome of different rearing systems while maintaining production and profits at a high level [1]. Motivational interviewing has been used to encourage farmers to take ownership of hens’ injurious pecking issues under different housing systems, which contributed to improving both the sustainability of egg production and hen welfare [2]. Cumulative mortality was found to be different between housing systems (free range system was supposed to have higher mortality than conventional cage system) through analysis of data from ten sources comprising 3851 flocks [3]. Rearing conditions are increasingly recognised as a critical factor in laying the foundations for subsequent behaviour, production, health, and welfare of hens.

Cage design is one of the crucial factors of rearing conditions contributing to the environment of hens and plays a vital role in affecting their well-being and production performance [4]. Decreased cage space has been reported to decrease the laying rate, egg quality, body weight, and feed efficiency [5,6]. The lack of movement in small cages could cause metabolic disorders, high rates of disuse osteoporosis, and the birds experience severe frustration due to the prevention of normal behaviours [7]. Other problems associated with a low space allowance include increased heterophil: lymphocyte ratios and increased serum malondialdehyde levels of birds, which indicated that crowding induced physiological and oxidative stress [8,9]. However, the dynamics of space utilisation are changed when the cage size is increased to accommodate larger flocks, increasing the overall amount of relative space accessible to individual hens [10], which could provide more opportunities for them to perform foraging, wing stretching, and leg stretching [11]. Therefore, increasing the cage size may be beneficial to the production and health of laying hens.

Several studies reported that chickens reared in large furnished cages laid more and bigger eggs with heavier eggshell than those in small conventional cages [12,13], and had higher abundance of *Faecalibacterium* and *Butyricimonas* in caecum [14]. Moreover, Meng et al. [6] found that hens from large cages had higher eggshell strength and eggshell weight than those from small cages. Li et al. [15] found that large and medium cages were more conducive to the growth of chicks as compared to small cages, and thus decided to conduct research on increasing cage size to improve pullet growth. It was previously reported [16] that birds in large cages had better body mass and serum biochemical indices (lower serum TG and T-CH concentrations) than those in small cages under the same nutritional conditions. However, aggression and movement were found to be more frequent in large cages than in small cages under a hot environment, which led to lower egg production and feed conversion ratios [17].

According to the aforementioned studies, cage size has a significant impact on the health and productivity of hens, and thus it is necessary to determine the extent of its contribution. In this study, we hypothesised that different-sized cages might have strong effects on laying hens due to their different bottom dimensions (in the horizontal plane) and relative space availabilities. Therefore, four different-sized cages, i.e., huge cages (HC), large cages (LC), medium cages (MC) and small cages (SC), were used under the same management conditions to investigate their effects on the productive performance, serum biochemical indices, and caecal microbiota composition of hens.

## 2. Materials and Methods

### 2.1. Animals and Management

The study was approved by the Institutional Animal Care and Use Committee (IACUC) of Anhui Academy of Agricultural Sciences under approval number A11-CS06. The experiment was conducted at Anhui Sundaily Village Ecological Food Co., Ltd. (Tongling, China). A total of 450 44-week-old healthy wing-tagged Roman laying hens with similar body weights (1762.87 g ± 172.36) were selected and randomly divided into four groups according to the different cage sizes: huge cage (HC, 0.6 m × 2.25 m × 0.4 m), large cage (LC, 0.6 m × 1.35 m × 0.4 m), medium cage (MC, 0.6 m × 0.9 m × 0.4 m), and small cage (SC, 0.6 m × 0.45 m × 0.4 m). There were no enrichment materials inside the cages. The stocking density across the four groups was identical (0.054 m^2^ per bird) throughout the experimental period (Table 1). A two-week preliminary trial was conducted for hens to adapt to the cages (as hens were raised in SC prior to the trial), and then the formal experiment was performed from 46 to 60 weeks of age. During the trial period, the average indoor temperature and relative humidity were controlled at 22 °C and 53%, respectively. All chickens were subjected to a light/dark cycle (16 h light: 8 h dark) and given free access to feed and water. Feed was supplied in troughs placed in front of each cage twice a day, and water was provided from nipple drinkers continuously. The experimental diets were formulated according to NRC (1994) recommendations for laying hens [18].

### 2.2. Production Performance

To determine the laying rate and average egg weight, eggs were collected and weighed on a daily basis per cage. The average egg weight was multiplied by the hen-day laying rate to determine the egg mass. To analyse production performance, the average daily feed intake (ADFI) and feed conversion ratio (FCR) were tracked and calculated. The ratio of feed intake per unit of egg mass was used to calculate the FCR. The body weight gain (BWG) of the birds was weighed on an empty stomach (after 12 h of feed withdrawal) every two weeks.

### 2.3. Serum Biochemical Parameters and Antibody Titres

Eighteen wing-tagged birds from each group (3 per replicate) after 12 h of feed withdrawal were selected for blood sampling and designated for the determination of antibody titres against the avian influenza viruses H5N1, H7N9, and H9N2 and against Newcastle disease virus (NDV). Two heparinised tubes were used to collect a 4 mL sample of blood from the hens’ wing veins (2 mL in each tube). After collection, samples were immediately put in an ice bath before transport to the lab for processing. Antibody titres were measured by enzyme-linked immunosorbent assay (ELISA) kits (Mlbio Biotech Co., Shanghai, China) according to the manufacturer’s protocol. Antibody titre data were logarithmically transformed (base 2) prior to analysis. At the end of the experiment (60 weeks of age), twelve wing-tagged birds from each group (2 per replicate) after 12 h of feed withdrawal were selected for blood sampling and designated for the determination of serum biochemical parameters, including triglycerides (TG), total cholesterol (T-CH), malondialdehyde (MDA), glutathione peroxidase (GSH-Px), superoxide dismutase (SOD), corticosterone (CORT), and immunoglobulin G (IgG). Blood serum was centrifuged at 4 °C for 10 min (3000× *g*) to separate the plasma, and then was stored at −20 °C until analysis. Commercial analytical kits (Sigma, Thermo Fisher Scientific, Shanghai, China) were used with an autoanalyser (Hitachi Ltd., Tokyo, Japan) to determine the concentrations of these parameters.

### 2.4. Sample Collection

Six birds from each group (one from each replication) were selected at the age of 60 weeks to be used in the collection of intestinal content. Birds were sacrificed by CO_2_ suffocation (exposure to 50% CO_2_ for 5 min) and then intestines were removed aseptically from the abdominal cavity. Caecal contents were gently squeezed into 2 mL cryopreservation tubes and frozen instantly at −80 °C for further analysis.

### 2.5. Bacterial DNA Extraction and PCR Amplification of 16S rRNA

HiPure Stool DNA kits (Magen, Guangzhou, China) were used to extract bacterial genomic DNA according to the manufacturer’s protocols. The V3-V4 regions of 16S rDNA were PCR-amplified from microbial genomic DNA using the universal primers V341F (5′-CCTACGGGNGGCWGCAG-3′) and V806R (5′-GGACTACHVGGGTATCTAAT-3′). PCR was performed in a 20 μL reaction system containing 0.8 μL of each primer, 10 ng of template DNA, 4 μL of 5 × FastPfu buffer, 2 μL of 2.5 mM dNTPs, and 0.4 μL of FastPfu polymerase. The PCR conditions were as follows: 95 °C for 2 min, followed by 30 cycles of 95 °C for 30 s, 61 °C for 30 s, and 72 °C for 60 s, with a final extension of 72 °C for 10 min. Amplicons were pooled, purified, and then quantified using a NanoDrop 2000 UV-VIS instrument (Thermo Scientific, Wilmington, DE, USA).

### 2.6. 16S rRNA Sequencing and Bioinformatics Analysis

Next-generation sequencing was performed with an Illumina HiSeq 2500 PE250 system by Gene Denovo Biotechnology Co., Ltd. (Guangzhou, China) to investigate the bacterial community. The raw sequence data were processed and analysed with the QIIME software package [19]. Then, sequences with a threshold of 97% similarity were clustered into operational taxonomic units (OTUs) using the UPARSE pipeline (version 9.2.64) [20]. These OTUs were used for diversity (Shannon and Simpson indices), richness (ACE and Chao indices), and rarefaction curve analyses using MOTHUR [21]. Taxonomic assignments of OTUs that reached the 97% similarity level were made using QIIME by comparison with the SILVA databases (http://www. arb-silva.de, accessed on 3 July 2022) [22]. Venn analysis of OTUs between groups was performed in R project VennDiagram package (version 1.6.16) [23]. Non-metric multidimensional scaling (NMDS) plots of sequence-read abundances were generated with the Vegan package in R software. A test for statistically significant group differences was performed using the non-parametric ANOSIM procedure.

### 2.7. Statistical Analysis

The data were subjected to analysis of variance (ANOVA) using the PROC GLM procedure in SAS statistics software version 9.3 (SAS Institute Inc., Cary, NC, USA). The cage size and age were the main effects. The productive traits, serum parameters, and antibody titres were analysed using the mixed model (PROC MIXED) procedure for repeated measurements, and the number of cages (replicates of each group) was determined as the random factor in the model. The significance of the differences between group means was tested using Tukey’s multiple comparison; *p* < 0.05 was considered significant. Data were expressed as the means ± standard error (SE). Differences in the relative abundances of microbial community compositions between groups were analysed with a two-tailed non-parametric Mann-Whitney U test. Correlations between the differential caecal bacterial genera and productive traits and serum biochemical indices were analysed using Spearman’s correlation analysis. Bonferroni’s correction was applied to reduce the probability of a type I error when calculating the correlation coefficients [24], and *p* < 0.002 was considered significant.

## 3. Results

### 3.1. Production Performance

The effects of cage size on productive performance are presented in Table 2 and Figure 1. The highest laying rate of birds was found in the MC group (*p* < 0.05), while the lowest laying rate was found in the HC group (*p* < 0.05). Significant differences were found for the laying rate among the four groups from 52 to 58 weeks of age (*p* < 0.05), which were ranked as MC > LC > SC > HC. The FCR of birds in LC and MC was significantly lower than those in HC and SC from 50 to 60 weeks of age (*p* < 0.05), except at 54 weeks of age. Birds raised in SC had the highest BWG, which was approximately 2 times higher than that of birds raised in LC and MC and 1.6 times higher than that of birds raised in HC (*p* < 0.05). There was no significant difference for egg weight among the four groups (*p* > 0.05).

### 3.2. Immune Response

The effects of cage size on serum antibody titres of H5N1, H7N9, and H9N2 and against NDV are shown in Figure 2. The antibody titres against H5N1 significantly increased in the birds from LC and MC compared with those from HC and SC at 54 and 57 weeks of age (*p* < 0.05). Additionally, the levels of antibody titres against H9N2 were significantly higher in birds from LC and MC than in birds from HC and SC at 57 weeks of age. No significant differences were found in H7N9 and NDV antibody titres among the four groups.

### 3.3. Serum Biochemical Parameters

The effects of cage size on the serum biochemical indices of birds are presented in Table 3. With increasing cage size, the serum T-CH and TG levels of birds decreased and the serum GSH-Px activity improved, where birds raised in HC had the lowest serum TG and T-CH and the highest GSH-Px activity. Serum SOD and IgG levels were slightly increased in the HC, LC, and MC groups as compared to the SC group (*p* > 0.05). No significant differences in serum MDA or CORT concentration were found among the four groups (*p* > 0.05).

### 3.4. Diversity Analysis of the Microbiota in the Caecum

Alpha diversity (effective reads, Chao1, Ace, Shannon, and Simpson indices) was determined to describe species richness and evenness. A total of 426,029.66 effective reads were obtained from 24 caecal content samples using 16S rRNA gene sequencing, with average lengths of 454.63, 453.92, 451.70, and 455.45 reads for the HC, LC, MC, and SC groups, respectively (Table 4). With 97% sequence similarity, a total of 5430.32 bacterial OTUs were found, with an average number of 1375.33, 1380.33, 1385.33, and 1289.33 for the HC, LC, MC, and SC groups, respectively. The Ace and Chao methods were used to estimate bacterial richness indices based on OTUs, while the Simpson and Shannon methods were used to calculate bacterial diversity indices. The OTUs’ produced rarefaction curves, which indicated that good sample coverage was attained across all groupings (Figure 3). Birds from different-sized cages had different bacterial OTUs in caecal contents (Figure 4). These four groups shared 676 bacterial OTUs, whereas the HC, LC, MC, and SC had unique sequences for 102, 124, 127, and 148 bacterial OTUs, respectively.

Beta diversity was used to assess variations in species diversity between multiple samples. The NMDS results showed the difference in microorganism distributions among the four groups. The NMDS of the caecal microbiota community structure of the four groups is shown in Figure 5a, which shows that the differences among groups were greater than those within groups, indicating that the grouping was effective. The sample points from the SC group showed distinct distances from other groups, while there were shorter distances among the HC, LC, and MC sample points. Using ANOSIM, the four groups’ caecal microbiota compositions were compared, and the results (Figure 5b) demonstrated significant differentiation (*R* = 0.127, *p* = 0.013).

### 3.5. Composition Analysis of the Microbiota in the Caecum

All sequences were categorised from phylum to species based on the SILVA taxonomy database and the RDP Classifier14 analysis tool. A total of 29 different phyla were detected in the samples. The dominant phyla present in the caecal contents were *Bacteroidetes*, *Firmicutes*, *Actinobacteria*, *Proteobacteria*, and *Euryarchaeota* for the four groups (Figure 6a, Figure 7a and Appendix A). The relative abundance of *Bacteroidetes* was significantly lower in the MC group (34.54%), while the relative abundance of *Firmicutes* (52.35%) was significantly higher in the MC group than in the other groups (*p* < 0.05). *Euryarchaeota* was significantly more abundant in the SC group and was significantly less abundant in the MC group than in the HC and LC groups. Other dominant bacterial phyla did not significantly differ among the four groups (*p* > 0.05).

At the genus level, a total of 301 different genera were detected, and the most abundant bacteria were Bacteroides, Lactobacillus, Faecalibacterium, Prevotellaceae_UCG-001, Rikenellaceae_RC9_gut_group, Ruminococcus_torques_group, Desulfovibrio, Olsenella, Methanobrevibacter, and Alloprevotella (Figure 6b and Figure 7b and Appendix A). The relative abundance of Bacteroides was the lowest in the MC group (12.04%) compared with the other groups (*p* < 0.05). Lactobacillus was significantly less abundant in the LC (6.01%) and MC (5.91%) groups than in the HC and SC groups (*p* < 0.05). Faecalibacterium was significantly more abundant in the MC group (15.99%), reaching an approximately 4 times higher abundance than that of the HC and LC groups (*p* < 0.05), and was significantly less abundant in the SC group (*p* < 0.05). The relative abundance of Methanobrevibacter was significantly higher in the SC group than in the other groups (*p* < 0.05). No significant difference was found for other dominant genera in the caecum among the four groups (*p* > 0.05).

### 3.6. Correlation Analysis of Differentially Detected Bacterial Genera with Productive Performance and Serum Biochemical Indices

Spearman’s correlation analysis was performed based on the relative abundance of the above differential bacterial genera and productive performance and some serum biochemical indices (Table 5). *Lactobacillus* was significantly positively correlated with body weight (*R* = 0.350, *p* = 0.0009) but was negatively correlated with the laying rate and FCR. *Faecalibacterium* was positively correlated with the laying rate and FCR and was significantly positively correlated with the serum GSH-Px concentration (*R* = 0.214, *p* = 0.0017).

## 4. Discussion

Cage size is a crucial external factor that affects both growth and health of birds, and plays an important role in determining its performance and well-being [25]. The current results revealed that different cage sizes affected the productive performance of laying hens, although not all traits were significantly influenced. The highest laying rate and lowest FCR in birds raised in MC indicated positive effects on egg production and feed efficiency of birds compared to SC. Larger cages may offer a greater relative activity area and change the dynamics of space utilisation [4], which may improve laying performance compared to smaller cages, despite maintaining stocking densities equally across the four groups. However, no significant effects were found for production performance caused by HC and LC compared to SC, which suggested that cage size is not the larger the better. Similarly, Meng et al. [6] reported that hens in large furnished cages tended to have lower daily feed intake and egg production than those in small furnished cages. The highest ADFI and BWG in found in birds raised in SC indicated that most of the feed consumption was converted to meat but not egg, which led to the lowest feed-to-egg ratio compared to other groups. Smaller space allowance in small cages limited hens’ innate behaviours, such as foraging, wing stretching, and feather pecking. Thus, hens stayed inactive most of the time after feed intake, possibly resulting in abdominal fat deposition, which in turn, may increase the body weight and reduce the egg production [26]. This speculation needs further investigation.

Serum biochemistry parameters reflect the physiological and metabolic condition of birds and are affected by various factors, among which the space allowance is one of the most important [27,28]. T-CH and TG are related to lipid metabolism, which is important for determining the health status. The antioxidant state of animals is often represented by the antioxidant indices GSH-Px and SOD [29]. In the present study, there was a downwards trend in the serum TG and T-CH concentrations in birds with increasing cage size, because larger cages could provide adequate space for locomotion and basic movements of hens, while exercise can briefly lower blood lipids [30]. This finding was similar to that of our previous study [16], in which birds in large cages had lower concentrations of T-CH and TG than those in small cages under the same nutritional conditions. Simsek et al. [9] found that a small space allowance reduced the serum GSH-Px level of Ross-308 broiler chickens, while Li et al. [15] reported that the plasma GSH-Px and SOD levels of Jinghong chickens in small cages were higher than those in large and medium cages. In this study, a significant increase in GSH-Px activity and higher SOD and MDA levels were found in the MC, LC, and HC groups than in the SC group, which showed the superiority of the large cage in oxidation resistance, indicating a better hen physiological state. The inconsistent results among studies could be attributed to the different dietary nutrient levels and chicken breeds.

It has been suggested that the immune performance of poultry may be influenced by cage sizes [31,32]. Li et al. [15] reported that birds in large cages had higher antibody titres of H9N2 and H5N1 than those in medium and small cages. Similarly, although there were no significant differences in antibody titres among the four groups throughout most of the experimental period, the higher level of antibody titres against H5N1 (at 54 and 57 weeks of age) and H9N2 (at 57 weeks of age) in the LC and MC groups than in the HC and SC groups indicated that an appropriate increase in cage size could benefit bird health and maintain higher serum antibody titres against H5N1 and H9N2.

The caecum Is a dynamic ecosystem with a diverse microbiome, and its microbial composition is influenced by various factors, including age, diet, and rearing conditions [33]. In the present study, we allocated laying hens to four different-sized cages and kept other factors identical to investigate their effects on the caecal microbiota. The results of 16S rRNA profiling revealed that cage size affected the composition of caecal microbiota in hens. At the phylum level, the lowest abundances of *Bacteroidetes* and *Euryarchaeota* and the highest abundance of *Firmicutes* were found in the MC groups, while the lowest abundance of *Firmicutes* and the highest abundance of *Euryarchaeota* were found in the SC group. In poultry production, bacteria related to productivity and metabolism mainly include the phyla *Bacteroidetes*, *Firmicutes*, and *Proteobacteria* [34]. *Bacteroidetes* was reported to participate in a variety of metabolic processes, such as the utilisation of nitrogenous substances, digestion of carbohydrates, and maintenance of intestinal microecological balance [35]. A higher ratio of *Firmicutes*/*Bacteroidetes* in the caecal microbiota was reported to be associated with higher feed energy utilisation in Dagu chickens [36]. Consistent with the results of the present study, laying hens from MC had a higher laying rate and feed conversion efficiency with a higher *Firmicutes*/*Bacteroidetes* ratio than those from SC, which had a lower *Firmicutes*/*Bacteroidetes* ratio.

The differences in caecal microbes of laying hens from the four different-size cages are likely to be caused by the differences of space allowance and environmental stress. At the level of the most abundant genera, the abundance of *Lactobacillus* was higher in hens from HC than in hens from LC and MC, while the abundance of *Bacteroides* was the lowest in birds from MC. *Lactobacillus* is considered an important probiotic in the intestine of animals and is conductive to digestion and immunity [37], and was reported to be more highly abundant in the caecum of chickens who exhibited better health status [38]. The higher abundance of *Lactobacillus* in hens from HC may related to the larger space allowance and lower crowding stress, while oxidative stress could be induced by the crowding in smaller cages to promote pathogenic bacteria and reduce *Lactobacillus* [39,40]. It is believed that stressful conditions such as crowding could disrupt the microbial ecology of the bird’s intestine, thereby causing dysbiosis [41]. It has been reported that high stocking density may lead to increased dust and airborne pathogens [42], and *Lactobacilli* count in the intestinal contents of birds was negatively influenced by high stocking density [43]. In addition, hens in small cages lacked adequate space for movement and experienced inability to experience positive affective states, while social behaviours (such as congregating, feather pecking, and competition for feed) may occur more frequently in larger cages as birds were establishing their social hierarchy, possibly leading to birds ingesting more bacteria, etc., off of the feathers of other birds and consequently changed the caecal microbiota compositions. Spearman’s correlations showed that the abundance of *Bacteroides* and *Lactobacillus* was weakly correlated with the laying rate, FCR, and serum parameters, but *Lactobacillus* was significantly positively correlated with body weight (*R* = 0.350, *p* = 0.0009). This was in accordance with the results that the body weights were higher in birds from HC and LC with higher abundances of *Lactobacillus*.

Notably, the abundance of *Faecalibacterium* was significantly higher in the MC group, and the abundance of *Methanobrevibacter* was significantly higher in the SC group than in the other groups. *Faecalibacterium* are the primary bacteria involved in the production of short-chain fatty acids and play a role in resistance to colonisation by gut pathogenic microorganisms [44]. Caecal *Faecalibacterium* was positively correlated with the laying rate and FCR and significantly positively correlated with serum GSH-Px (*R* = 0.214, *p* = 0.0017), which suggested that the high abundance of *Faecalibacterium* in the MC group may contribute to the production performance and antioxidant capacity of hens. *Methanobrevibacter*, which belongs to the *Euryarchaeota* phylum, is a common and important methanogenic taxon that primarily inhabiting in the chicken caecum. Chickens with fewer *Methanobrevibacter* were found to have significantly lower abdominal fat content than those with a higher abundance of *Methanobrevibacter* [45]. Caecal *Methanobrevibacter* was negatively correlated with laying rate and FCR and positively correlated with serum TG and T-CH. Although the correlation was not significant, it is reasonable to hypothesise that the relatively poor laying performance and lipid metabolism may be related to the higher abundance of *Methanobrevibacter.*

## 5. Conclusions

In conclusion, large cage size has some positive effects on the production performance, lipid metabolism, and antioxidant capacity of Roman chickens. MC were superior to HC and LC in improving feed conversion efficiency and caecal microflora composition compared to the SC. It was suggested that an appropriate increase in cage size is beneficial to the production and health of laying hens. Further studies are needed to explore the optimum cage size during other laying periods in different chicken breeds.

## Figures and Tables

**Figure 1 animals-13-00266-f001:**
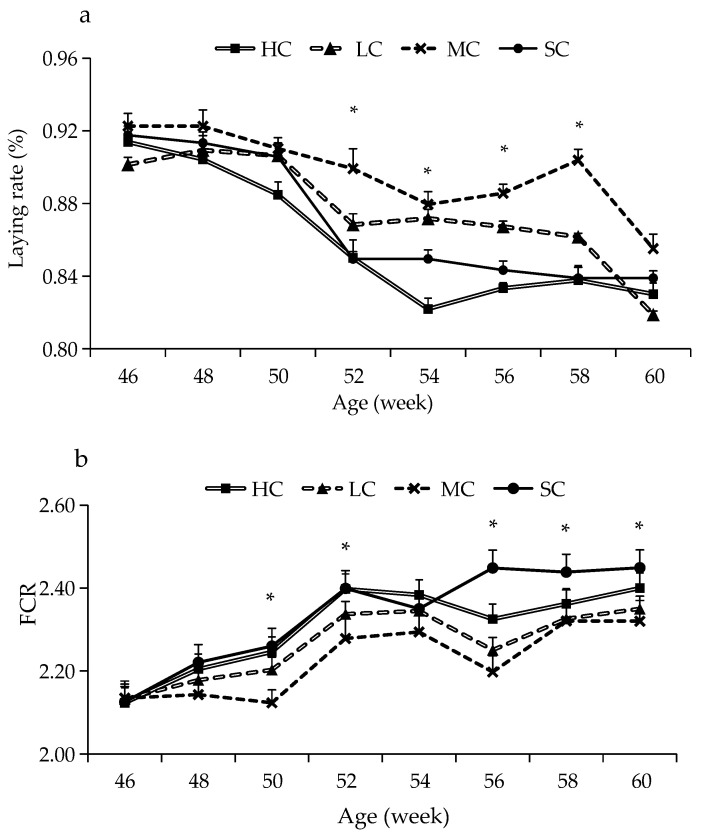
Laying rate (**a**) and feed conversion ratio (FCR) (**b**) of laying hens reared in different cage size groups from 46 to 60 weeks of age. HC, huge cage; LC, large cage; MC, medium cage; SC, small cage. * Within each time, means with an asterisk superscript are significantly different (*p* < 0.05).

**Figure 2 animals-13-00266-f002:**
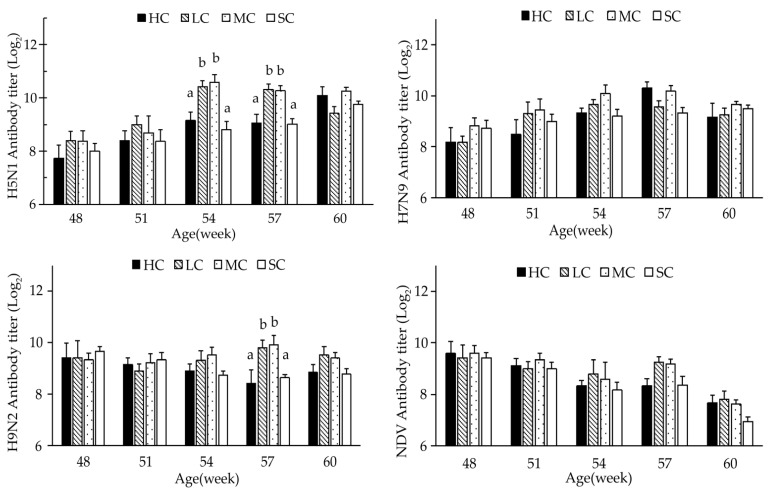
Antibody titre of birds reared in different cage size groups from 46 to 60 weeks of age. HC, huge cage; LC, large cage; MC, medium cage; SC, small cage. ^a,b^ Means within a week with different letters are significantly different (*p* < 0.05).

**Figure 3 animals-13-00266-f003:**
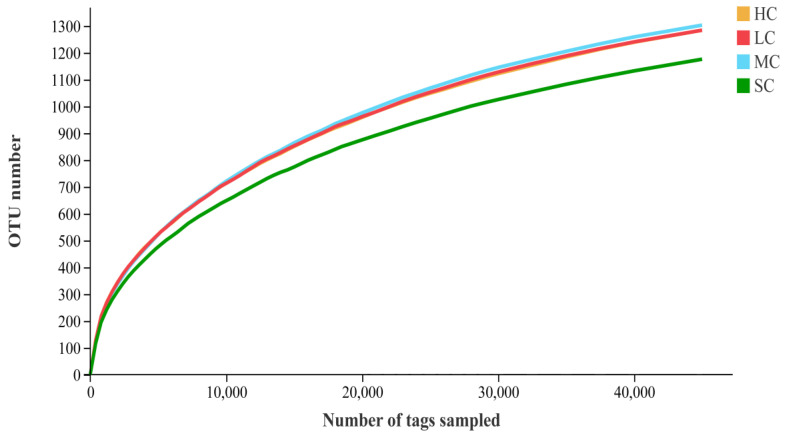
Rarefaction curves of the HC, LC, MC, and SC groups. OTU, operational taxonomic unit; HC, huge cage; LC, large cage; MC, medium cage; SC, small cage.

**Figure 4 animals-13-00266-f004:**
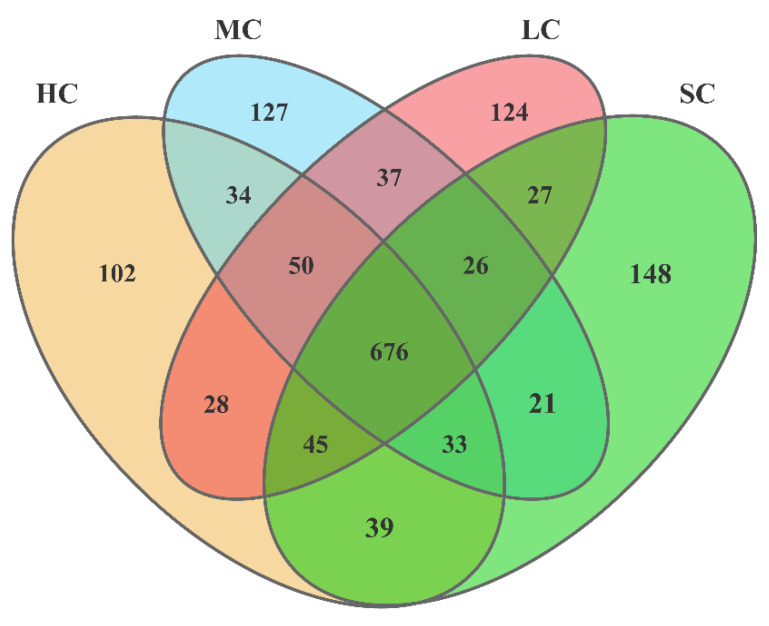
Flower plots of the caecal microbiota of laying hens reared in HC, LC, MC, and SC (based on OTUs). Each circle in the Venn diagram corresponds to a group that is represented by the same colour. The numbers in the overlapping areas indicate the number of bacterial OTUs shared between the respective groups. The numbers in the individual areas indicate the number of bacterial OTUs unique to that group. HC, huge cage; LC, large cage; MC, medium cage; SC, small cage.

**Figure 5 animals-13-00266-f005:**
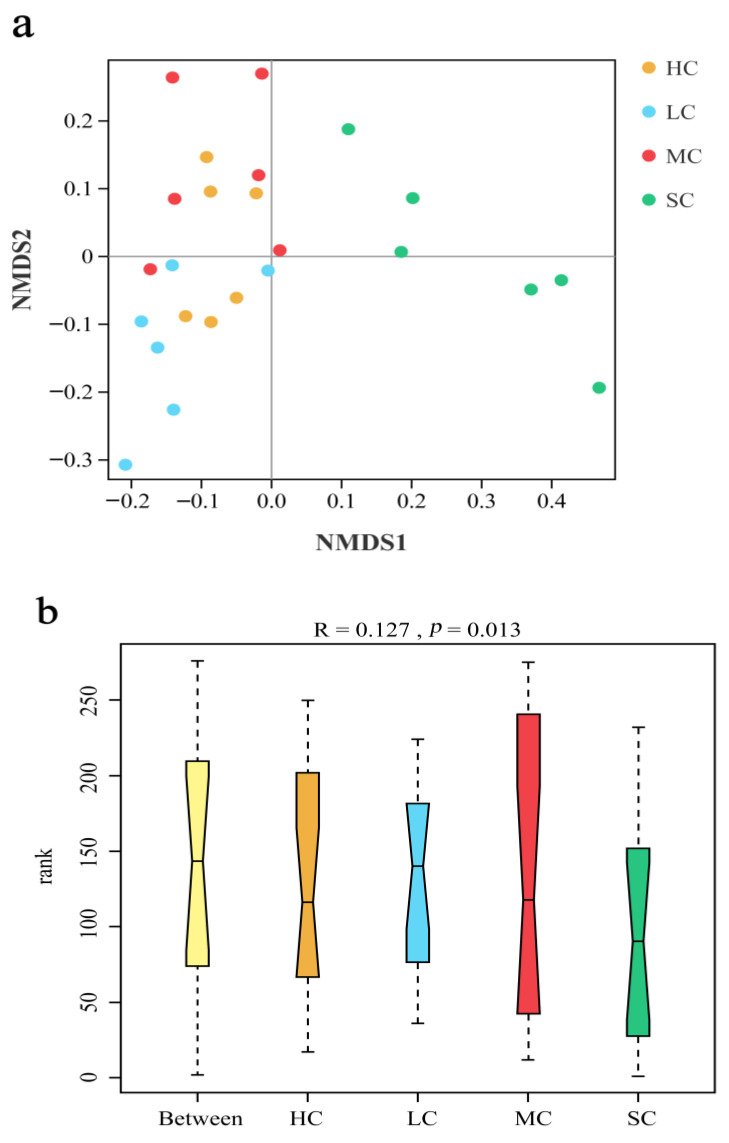
Non-metric multidimensional scaling (NMDS) ordination (**a**) and ANOSIM (**b**) of caecal microbiota (based on the unweighted UniFrac distance) in birds from HC, LC, MC, and SC. NMDS1 and NMDS2 on the *x*- and *y*-axes represent two principal discrepancy components among groups. Points represent samples. Samples in the same group share the same colour. HC, huge cage; LC, large cage; MC, medium cage; SC, small cage.

**Figure 6 animals-13-00266-f006:**
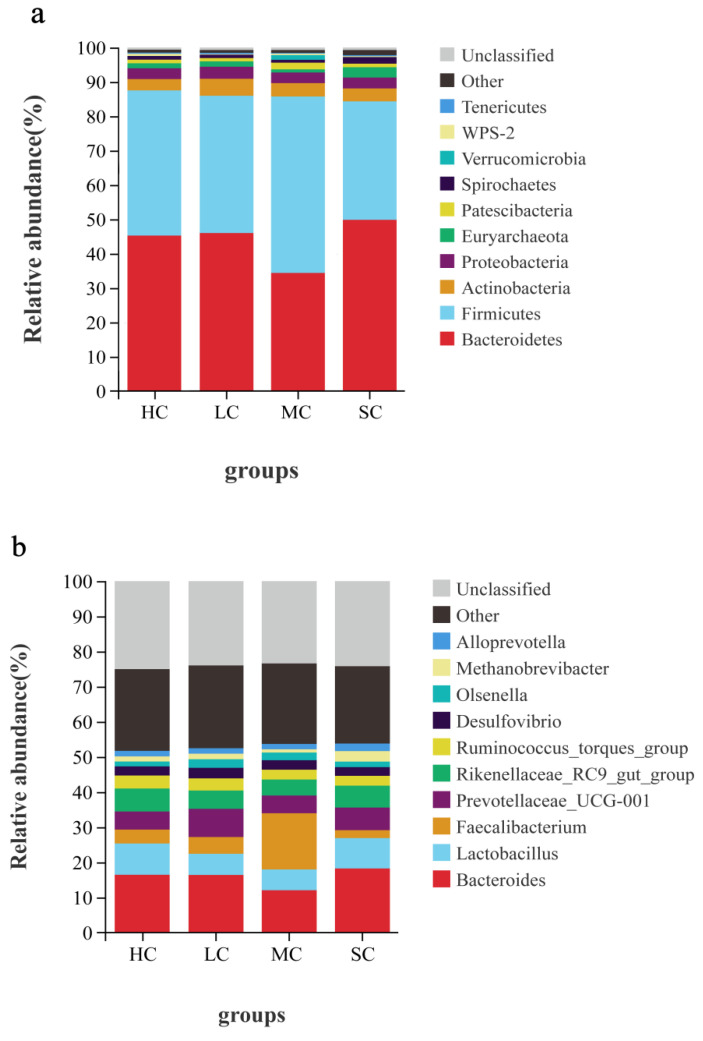
Relative abundance of caecal microbiome at the phylum (**a**) and genus (**b**) levels in laying hens reared in HC, LC, MC, and SC. HC, huge cage; LC, large cage; MC, medium cage; SC, small cage.

**Figure 7 animals-13-00266-f007:**
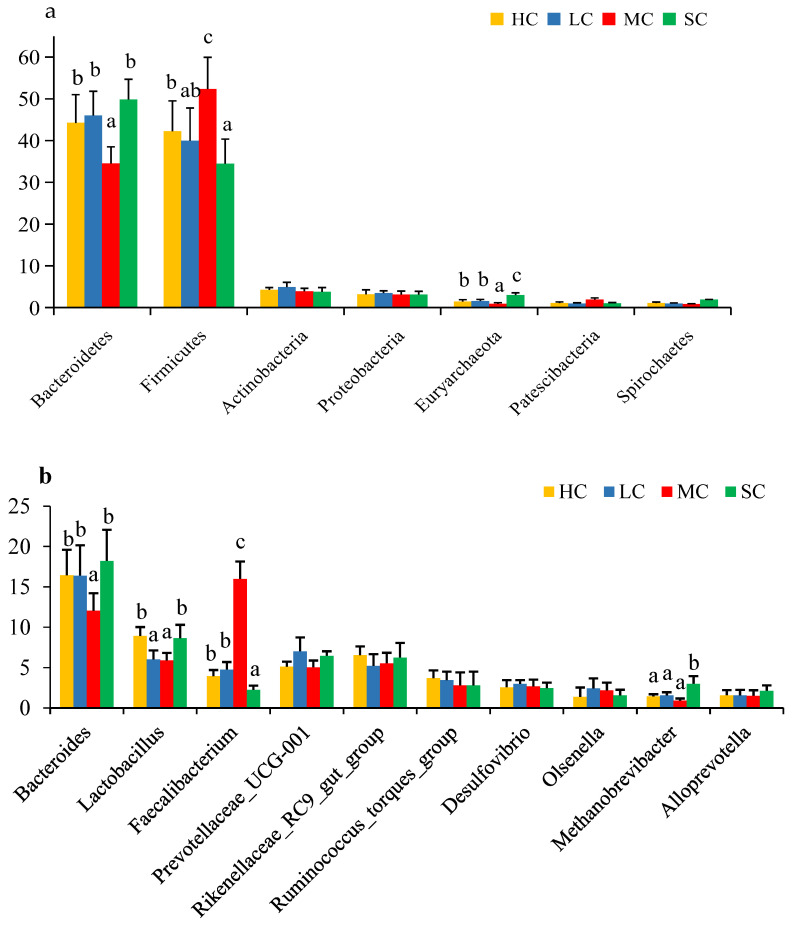
Relative abundance (% reads) of (**a**) the most dominant phylum and (**b**) the most dominant genus in the caecal microbiome of birds from in HC, LC, MC, and SC. Error bars represent the SE of samples. Boxes with different letters above the error bars are significantly different at *p* < 0.05. HC, huge cage; LC, large cage; MC, medium cage; SC, small cage.

**Table 1 animals-13-00266-t001:** Parameters of the experimental cages.

Item	HC	LC	MC	SC
Cage dimension (length × width × height, m)	0.6 × 2.25 × 0.4	0.6 × 1.35 × 0.4	0.6 × 0.9 × 0.4	0.6 × 0.45 × 0.4
Base area of cage (m^2^)	1.35	0.81	0.54	0.27
Birds/cage	25	15	10	5
Number of cages (replicates)	6	6	6	6 × 5
Total number of birds	150	90	60	150
Stocking density (m^2^ per bird)	0.054			

HC, huge cage; LC, large cage; MC, medium cage; SC, small cage.

**Table 2 animals-13-00266-t002:** Effects of cage size on the overall production performance of laying hens from 46 to 60 weeks of age.

Group	BWG (g)	Laying Rate (%)	ADFI (g)	Egg Weight (g)	FCR
HC	285.65 ± 10.78 ^a^	85.95 ± 1.08 ^a^	119.29 ± 0.86 ^ab^	61.95 ± 1.05	2.31 ± 0.05 ^b^
LC	227.73 ± 7.12 ^a^	87.56 ± 0.87 ^bc^	118.91 ± 0.98 ^a^	61.75 ± 0.97	2.27 ± 0.04 ^ab^
MC	202.24 ± 8.28 ^a^	89.73 ± 0.96 ^c^	117.55 ± 1.01 ^a^	61.62 ± 0.99	2.23 ± 0.04 ^a^
SC	454.99 ± 17.12 ^b^	86.96 ± 1.03 ^ab^	122.62 ± 0.81 ^b^	61.66 ± 1.41	2.33 ± 0.05 ^b^

^a–c^ Means with different superscripts within each column are significantly different (*p* < 0.05). HC, huge cage; LC, large cage; MC, medium cage; SC, small cage; BWG, body weight gain; ADFI, average daily feed intake; FCR, feed conversion ratio.

**Table 3 animals-13-00266-t003:** Effects of cage size on the serum biochemical indices of laying hens.

Item	HC (n = 12)	LC (n = 12)	MC (n = 12)	SC (n = 12)
TG (μmol/mL)	337.06 ± 5.12 ^a^	359.08 ± 5.61 ^b^	362.68 ± 4.91 ^b^	382.82 ± 4.99 ^c^
T-CH (μmol/L)	900.01 ± 13.13 ^a^	920.65 ± 13.82 ^ab^	955.35 ± 10.41 ^b^	998.09 ± 9.78 ^c^
MDA (nmol/mL)	15.72 ± 0.44	15.51 ± 0.58	15.21 ± 0.69	14.38 ± 0.51
GSH-Px (IU/L)	349.67 ± 4.38 ^c^	343.65 ± 3.58 ^c^	322.95 ± 3.22 ^b^	298.61 ± 3.17 ^a^
SOD (U/mL)	16.51 ± 0.51	16.19 ± 0.26	16.30 ± 0.31	15.78 ± 0.28
CORT (μg/L)	189.78 ± 4.24	193.30 ± 4.08	174.16 ± 3.31	181.95 ± 4.35
IgG (μg/mL)	1036.48 ± 21.33	1056.51 ± 24.67	1077.61 ± 27.68	1033.24 ± 22.26

^a–c^ Means with different superscripts within each line are significantly different (*p* < 0.05). HC, huge cage; LC, Large cage; MC, medium cage; SC, small cage; TG, triglycerides; T-CH, total cholesterol; MDA, malondialdehyde; GSH-Px, glutathione peroxidase; SOD, superoxide dismutase; CORT, corticosterone; IgG, immunoglobulin G.

**Table 4 animals-13-00266-t004:** Diversity estimation of the 16S rDNA gene libraries of the caecal microbiota in laying hens from different groups ^1^.

Group	Effective Reads	Average Length	OTUs	Simpson	Shannon	ACE	Chao	Good’s Coverage
HC (n = 6)	106337.50 ± 4038.45	454.63 ± 1.37	1375.33 ± 121.12	0.98 ± 0.00	7.23 ± 0.17	1576.16 ± 106.65	1494.55 ± 99.64	0.99 ± 0.01
LC (n = 6)	106424.83 ± 4757.15	453.92 ± 2.03	1380.33 ± 107.79	0.98 ± 0.00	7.24 ± 0.15	1587.90 ± 110.42	1504.42 ± 98.98	0.99 ± 0.00
MC (n = 6)	108906.50 ± 2699.38	451.70 ± 3.21	1385.33 ± 121.63	0.98 ± 0.01	7.13 ± 0.19	1587.68 ± 139.33	1505.09 ± 134.92	0.99 ± 0.01
SC (n = 6)	104360.83 ± 4051.66	455.45 ± 1.35	1289.33 ± 90.83	0.97 ± 0.02	6.72 ± 0.42	1493.62 ± 102.51	1415.16 ± 93.99	0.99 ± 0.00

^1^ Operational taxonomic units (OTUs) were defined at 3% dissimilarity. Richness estimators (ACE and Chao) and diversity indices (Simpson and Shannon) were calculated. HC, huge cage; LC, large cage; MC, medium cage; SC, small cage.

**Table 5 animals-13-00266-t005:** Correlations of differentially detected caecal bacterial genera with productive performance and serum biochemical indices of the four groups.

Genus	Body Weight	Laying Rate	FCR	Serum TG	Serum T-CH	Serum GSH-Px
*Bacteroides*	0.013	0.150	0.032	0.026	−0.211	0.131
*Lactobacillus*	0.350 *	0.138	0.178	−0.068	−0.138	0.043
*Faecalibacterium*	0.070	0.282	0.269	−0.091	−0.130	0.214 *
*Methanobrevibacter*	0.022	−0.245	−0.226	0.072	0.069	−0.086

* Correlation analysis was performed along with a Bonferroni significance correction and was considered significant at *p* < 0.002. FCR, feed conversion ratio; TG, triglycerides; T-CH, total cholesterol; GSH-Px, glutathione peroxidase.

## Data Availability

The raw data of 16S rRNA gene presented in the study are available in NCBI repository under the accession No: PRJNA893087.

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
