# Peer review of "Effects of Different-Sized Cages on the Production Performance, Serum Parameters, and Caecal Microbiota Composition of Laying Hens"

_animals, 2023, doi:10.3390/ani13020266_

Round 1

Reviewer 2 Report

General comments:

the authors investigate the effect of different-sized cages on performance, serum parameters and caecal microbiota of laying hens. Despite this, the manuscript is of interest for Animals’ readership, as the laying hens’ welfare (especially in cage production systems) was perceived as highly relevant by consumers. For this reason, authors need to emphasize this topic, especially since there is growing recognition that rearing is a key stage, laying the foundations for subsequent behaviour, production, health, and welfare. Some useful information can be found in: https://doi.org/10.3390/IECA2020-08830 ;

https://doi.org/10.1371/journal.pone.0146394 ;

https://doi.org/10.3390/ani12182307 ;

https://doi.org/10.7120/09627286.22.1.057 ;

that I suggest to include as reference in the introduction.

The text is fluently written, the approach used is clearly presented, and includes only minor typos, and it is easy to read. I have only minor comments for the authors to clarify before I can recommend this manuscript for publication. I recommend to include the pullet rearing conditions and cage characteristics (e.g., the presence of enrichments).

Apart from this, I only have specific comments:

L26: delete were

L188 and further in the manuscript: Large cage

Reviewer 3 Report

Thank you to the authors for submitting their interesting study of evaluating the effect of cage size on various outcomes such as productivity, antibody titers, serum biochemical parameters, and cecal microbiota. Upon review, I believe that there are two major considerations for the manuscript: 1) statistical analysis and 2) rationale for the metrics. More details for each are provided below with corresponding line numbers. In brief, the statistical analysis should consider adding the replicate as a random effect and using a Bonferroni’s correction with the correlations. The rationale for the metrics should be elaborated. Currently, it is not clear what the mechanism is for cage size to affect the various outcomes. Elaborating on the rationale in the introduction and in areas of the discussion would strengthen the hypotheses and flow of the manuscript.

Major revisions

Introduction: The introduction section needs more rationale about the relationship between cage size and productivity, antibody titers, serum biochemical parameters, and cecal microbiota. Why/how would cage size influence these components? Why/how would having more relative space influence these metrics? The introduction focuses mostly on bone health and growth, but bone health was not evaluated in the current study. Need more relevant background information to support the objective and hypothesis.

L 153: For the Statistical Analysis, it is not clear how the authors are accounting for the repeated measurements that were taken across ages for some of the outcomes (e.g., outcomes in Figures 1 and 2). Is a repeated measures ANOVA used? How were model assumptions evaluated? I believe that the authors should have included the replicates as a random effect in their model to account for possible variation introduced by the replicates. Furthermore, reporting means with their standard errors or even 95% confidence intervals would be more meaningful to understand the distribution of values for the populations rather than standard deviations. Also, the statistical models are based on differences between standard error, so reporting/displaying standard error instead of standard deviation would also improve interpretation of the figures.

L 150-152: The information about correlations needs to be moved to the Statistical Analysis section. Also, considering that 24 correlations were evaluated (Table 5), a Bonferroni’s correction should be applied to reduce the likelihood of Type I Error (Curtin & Schulz, 1998; doi: 10.1016/S0006-3223(98)00043-2).

L 332-334: Why was most of the feed converted to meat but not egg in SC? Please offer a proposed explanation of the mechanism and link to the literature.

L 344: “congregated together under natural conditions” does not make sense. A cage is not a “natural condition”. Congregating would imply that the birds huddled together and were inactive, but then you state that larger cages “stimulate the activities of birds”. This is vague. Please revise this interpretation. How does activity reduce blood lipids? What is the mechanism? What about the role of social behavior, such as feather pecking or competition for feed? These behaviors may have occurred in a larger group as birds were establishing their social hierarchy, possibly leading to birds ingesting more bacteria, etc. off of the feathers of other birds in the cage.

L 380: “environmental stress” is never explained. Please elaborate what are the ways in which the different cage sizes contribute to different levels of environmental stress. How is environmental stress linked to Lactobacillus? Please explain the mechanism.

L 391-393: Interesting result. Why do birds from the larger cage sizes have more Lactobacillus? The mechanism is not explained.

Minor revisions

Abstract: Please clarify in the abstract that stocking density was kept consistent, but cage size and number of birds/cage changed for each treatment.

L 51: “a at high level” typo; should be “at a high level”

L 70: “bottom areas” – I am not sure what this is referring to. I initially interpreted it as the surface material of the floor. Please be more specific.

L 92: In the Production Performance section, please include the number of eggs that were analyzed per replicate.

L 148: Please cite the R package and the version used. This can be obtained with R code: citation("Vegan")

packageVersion("Vegan")

L 188: “large cage” misspelled as “lagre cage”

L 329: “as larger as better” grammatical issue

Round 2

Reviewer 3 Report

Thank you to the authors for the thorough response to the suggested revisions. The manuscript has greatly improved. However, there is still one issue. The statement that a Bonferroni correction was applied was added to the methods in the manuscript, but I do not see evidence of the Bonferroni correction in the data. Typically, the p-value for significance would change when a Bonferroni correction is applied. For example, the standard alpha / significant p-value of 0.05 would be divided by the number of correlations to create a new level of significance that helps to reduce Type I Errors (0.05 / 24 correlations in this manuscript = 0.002). As a result, the p-value for a correlation to be significant would need to be equal to or less than 0.002. This p-value is more difficult to achieve, which is why it reduces Type I Errors with correlations. Currently, the significant alpha is still 0.05 and p-values are not displayed in the table. The authors either need to change the level of significance for the p-value to actually make the Bonferroni correction and reflect this in the methods and Table 5, or they need to remove the statement they added to the methods about the Bonferroni correction because it has not actually been applied. Statistically, adjusting the alpha to perform the Bonferroni correction appropriately would be preferred.

Author Response

Thank you for the careful guidance and sorry for our mistakes. Bonferroni’s correction was applied to reduce the probability of a type I error when calculating the correlation coefficients, and p <0.002 was considered significant. We have revised the description in the methods (Line 181-183) and Table 5.